# Spatial Equity of PM_2.5_ Pollution Exposures in High-Density Metropolitan Areas Based on Remote Sensing, LBS and GIS Data: A Case Study in Wuhan, China

**DOI:** 10.3390/ijerph191912671

**Published:** 2022-10-03

**Authors:** Zhuoran Shan, Hongfei Li, Haolan Pan, Man Yuan, Shen Xu

**Affiliations:** 1School of Architecture and Urban Planning, Huazhong University of Science and Technology, Wuhan 430074, China; 2Hubei Engineering and Technology Research Center of Urbanization, Wuhan 430074, China; 3The Key Laboratory of Urban Simulation for Ministry of Natural Resources, Wuhan 430074, China; 4Guangzhou Urban Planning Survey and Design Institute, Guangzhou 510060, China

**Keywords:** PM_2.5_ exposure, spatial equity, population differentiation, metropolitan areas

## Abstract

In-depth studies have been conducted on the risk of exposure to air pollution in urban residents, but most of them are static studies based on the population of residential units. Ignoring the real environmental dynamics during daily activity and mobility of individual residents makes it difficult to accurately estimate the level of air pollution exposure among residents and determine populations at higher risk of exposure. This paper uses the example of the Wuhan metropolitan area, high-precision air pollution, and population spatio-temporal dynamic distribution data, and applies geographically weighted regression models, bivariate LISA analysis, and Gini coefficients. The risk of air pollution exposure in elderly, low-age, and working-age communities in Wuhan was measured and the health equity within vulnerable groups such as the elderly and children was studied. We found that ignoring the spatio-temporal behavioral activities of residents underestimated the actual exposure hazard of PM_2.5_ to residents. The risk of air pollution exposure was higher for the elderly than for other age groups. Within the aging group, a few elderly people had a higher risk of pollution exposure. The high exposure risk communities of the elderly were mainly located in the central and sub-center areas of the city, with a continuous distribution characteristic. No significant difference was found in the exposure risk of children compared to the other populations, but a few children were particularly exposed to pollution. Children’s high-exposure communities were mainly located in suburban areas, with a discrete distribution. Compared with the traditional static PM_2.5_ exposure assessment, the dynamic assessment method proposed in this paper considers the high mobility of the urban population and air pollution. Thus, it can accurately reveal the actual risk of air pollution and identify areas and populations at high risk of air pollution, which in turn provides a scientific basis for proposing planning policies to reduce urban PM_2.5_ and improve urban spatial equity.

## 1. Introduction

During the past 40 years, China’s rapid urbanization and economic development have made tremendous progress. China’s urbanization rate has reached 65% and its gross domestic product has exceeded 100 trillion CNY. However, rapid urbanization has resulted in strained land resources, severe ecological degradation, and an immense increase in resource consumption, accompanied by various problems related to environmental pollution. Among them, the contradiction between problems of urban air pollution and high-quality urbanization development has become increasingly acute and complex. A large amount of epidemiological evidence has revealed a strong association between long-term exposure to air pollution (PM_2.5_, NO_2_, and SO_2_) and various diseases, including heart, respiratory, and cardiovascular disease, premature and low birth weight, and premature death [1,2,3,4,5,6]. According to a report by the Ministry of Ecological Environment of China, only 202 of the 337 cities monitored at the prefecture level and above monitored in 2020 met China’s ambient air quality standards (annual average: 35 µg/m^3^; 24-h average: 75 µg/m^3^), accounting for only 59.9% [7]. Along with the progress of urbanization in China, the population continues to flow into major urban agglomerations; however, problems such as uncoordinated development and high-intensity operation have made air pollution within urban agglomerations more apparent, turning them into “pollution clusters” [8]. According to the China Urban Agglomeration Integration Report, in 2020, there will be 910 million people living in 12 urban agglomerations in China, accounting for 64.6% of the total population. Widespread urban PM_2.5_ pollution seriously affects the health of residents and the sustainable development of cities. The report of the 19th Congress of the Communist Party of China pointed out that the main social contradiction in China is the contradiction between the people’s growing need for a better life and unbalanced and insufficient development. Health is the most basic condition required for humans to achieve a better life. China confers great importance on the health of its residents. The Outline of Health China 2030 Plan emphasizes the in-depth prevention and control of air pollutants, the establishment of a sound system of environmental and health monitoring, investigation and risk assessment, evaluation of the impact of environmental pollution on the health of the population, and the exploration of the establishment of a health risk assessment system. Therefore, accurate identification of areas and populations at high risk of air pollution exposure is a prerequisite for carrying out environmental optimization and protecting the health of residents and is the basis for achieving a better quality of life.

Research on air pollution and health has been a hot topic for scholars in different disciplines such as health geography, urban planning, environmental science, and medicine, and numerous researchers have studied urban pollution exposure and health risks (such as disease morbidity and mortality), urban residents’ spatial and temporal behavioral pollution exposure, and group health disparities and environmental justice. However, most of the current studies on group health disparities and environmental justice in air pollution exposure are mainly focused on western developed countries [9,10,11,12,13]. In addition, few studies have been conducted in China in recent years [14,15,16,17,18]. However, most of the existing research scales are single cities or urban clusters, and there is a lack of research on block and community scales within cities. In terms of research objects, population distribution is often characterized by census data, and the exposure risk of residents’ actual activities is replaced by the static air pollution exposure level of population living units, thereby ignoring the real dynamic environment in the process of individuals’ daily activities-mobility, which gives rise to the geographic context of uncertainty problem (UGCoP) [14]. In terms of air pollution exposure monitoring methods, low-cost air quality sensors are now widely used and have high accuracy, and they can be installed on buildings or carried by individuals [19,20,21,22]. However, their applicable spatial scales are currently mostly community scales and the population studied is mostly a small group of volunteers. The spatial scale of this study is Wuhan metropolitan area, and the population studied is tens of thousands, so the traditional national monitoring stations are still used as the data source. This paper aimed to accurately measure the actual risk of residents’ air pollution exposure and reveal the spatial pattern and equity of exposure risk in large cities with residents’ spatio-temporal activities. This paper reveals the spatial pattern of residents’ exposure risk in urban built environments based on hourly LBS (Location Based Services) cell phone signaling big data, high-precision air pollution data, and built environment walkability. At the same time, the differences in air pollution exposure among different groups and the derived environmental justice issues were studied in the context of the behavioral patterns of residents’ spatial and temporal activities to precisely identify the areas and populations at higher risk of air pollution exposure in the Wuhan metropolitan development area. This study provides a scientific basis for reducing the risk of air pollution exposure of urban residents and proposing more equitable and effective health policies.

## 2. Literature Review

### 2.1. Air Pollution Exposure Study Coupled with the Spatio-Temporal Behavior of Residents

Recently, air pollution exposure studies coupled with spatio-temporal behaviors of residents, focusing on the risk of air pollution exposure under individual or group spatio-temporal activities at the micro level, have become a hot research topic. Compared to the static assessment method, this method can reveal the air pollution exposure risk of residents with more accuracy and practicality. The research content includes the study of air pollution exposure under the daily behavioral pattern of the population, exposure based on individual spatial and temporal behavior trajectories, and exposure based on the dynamic distribution of the population.

Air pollution exposure studies based on the daily behavioral patterns of populations are carried out by categorizing the population based on the behavioral patterns of residents, followed by an investigation of the air pollution exposure levels of residents under different behavioral patterns. For example, Stefan Reis studied the air pollution exposure levels of UK residents at their workplace and residence and found that exposure to both NO_2_ and PM_2.5_ increased when occupational and residential activities were considered, compared to the exposure levels only at residences [9]. Subsequently, such studies extended the influence of the built environment such as urban spatial structure, walkability, and block scale on residents’ daily behavioral patterns to investigate their air pollution exposure levels under such influence: for example, urban street walkability affects residents’ choice of travel mode and the distance they travel, which in turn affects their exposure levels [15].

Air pollution exposure studies based on individual spatio-temporal behavioral trajectories are a type of air pollution exposure level measurement for individuals, usually based on GPS and PM_2.5_ real-time monitors, to measure the air pollution exposure level of individuals in their behavioral trajectories. Kwan Mei-po et al. used the activity trajectories of 117 volunteers in a highly polluted neighborhood in Beijing to accurately measure their exposure and found that residents were able to reduce their exposure risk through daily activities, but those with mobility impairments were always at a higher risk [16]. Using GPS data of residents’ trips and activity log survey data, Guo Wenbo et al. measured the differences in residents’ air pollution exposure levels under different modes of transportation and concluded that the motor vehicle travel rate was directly proportional to the air pollution exposure levels [17].

The study of air pollution exposure based on dynamic population distribution focuses on the uncertainty of group activity space, integrates the dynamic migration changes of population and air pollutants, and uses the actual space of residents’ activity/travel as the measurement unit to measure the actual pollution exposure level of the group [10,14]. In comparison with static assessment methods, dynamic assessment can measure the exposure risk of a group more accurately and practically. Bart Dewulf et al. measured the daily air pollution exposure of residents using the cell phone signaling data of approximately five million cell phone users in Belgium and showed an increase in exposure compared to static exposure [11]. Using residential OD (origin-destination survey) travel data to simulate residents’ commuting trajectories, Kwan Mei-po et al. used kriging interpolation to measure residents’ pollution exposure concentrations, and compared residents’ static pollution exposure concentrations to conclude that ignoring residents’ spatio-temporal activities can lead to incorrect assessment of exposure results [12]. Bin Chen et al. combined mobile phone location request (MPL) big data and PM_2.5_ observations from stations and introduced the population. The results showed that the risk of PM_2.5_ pollution exposure based on dynamic population distribution and census-based measurements deviated significantly [18]. Claudio Gariazzo et al. conducted a dynamic air pollution exposure assessment study using cell phone traffic data for an urban population in Rome, Italy and found that PM_2.5_ pollution exposure risk measured based on census data was significantly low [13].

### 2.2. Group Differences in Air Pollution Exposure and Environmental Justice Studies

Group differences in air pollution exposure are mainly based on the classification of populations based on socioeconomic attributes to study the levels of air pollution exposure in different populations and their influencing factors. The theoretical basis of such studies is that differences in social attributes affect the spatio-temporal behavior of residents, leading to differences in their air pollution exposure levels. That is, people with different social attributes choose different spatial locations and behavioral patterns due to their economic conditions and thinking patterns, and thus their air pollution exposure levels vary. The level of air pollution exposure of residents varies with age, income, household registration, gender, education level, and ethnicity [23,24].

Studies on air pollution exposure justice based on age characteristics have shown that the elderly and children bear a greater environmental burden. Pearce et al. reported that in the United Kingdom, the elderly and children produced less air pollution but were exposed to higher concentrations of air pollution, presenting environmental inequities [25]. A study by Jing Ma et al. in Beijing found that older people occupied an increasing proportion of areas with declining air quality and that environmental inequities increased over time [26].

Studies on the fairness of air pollution exposure based on income disparities have shown that low-income populations tend to be exposed to higher air pollution concentrations. Mitchell et al. studied NO_2_ pollution exposure levels of community residents in the United Kingdom and found that most community residents in communities with high exposure levels belonged to low-income groups [27]. Pearce et al. found that outdoor pollution levels in New Zealand were higher in socially deprived areas and settlements with a high proportion of low-income households [28]. Meir’s study in the United States found that low-income minorities found it difficult to purchase homes in clean areas, thereby increasing their exposure to pollution [29]. Marshall found that low-income households were more likely to be exposed to high levels of air pollution [30].

Other factors such as education level [31,32], the proportion of the migrant population [33], and race [34,35] have also influenced environmental injustice. Marshall’s findings suggested that PM_2.5_ pollution exposure was lower among highly educated people on the South Coast of California [30]. By investigating differences in exposure to environmental hazards and related health outcomes among Chinese residents, Juan Chen et al. found that migrant populations moving from rural to urban areas face more water pollution and higher levels of overall environmental hazards [33]. Through a study in Atlanta, Kwan Mei-po et al. found that pollution exposure levels were at high levels during the day for different races, but only white residents were able to reduce their nighttime pollution exposure by living in suburban areas away from busy roads [35].

### 2.3. Inadequacy of Existing Studies

In summary, dynamic air pollution exposure studies that couple the spatio-temporal behavior of residents have higher accuracy than static assessment methods. Environmental injustice is widespread worldwide, and residential air pollution exposure levels vary according to age, income, household registration, gender, education level, and race. However, most of the current studies focus on macro-scale such as individual cities or urban agglomerations, and micro-scale such as individual communities or residential blocks, and there is a lack of environmental equity exploration at the scale of urban communities. As a result, most of the research findings are qualitative and broad, and it is difficult to accurately identify urban areas and populations at high risk of air pollution. Over the years, the rapid urbanization and industrialization in China have placed a great burden on the ecological environment, and environmental injustice has intensified. To address the contradiction between people’s growing need for a better life and unbalanced and insufficient development, China emphasizes the establishment of a sound system of environment and health monitoring, investigation, and risk assessment, evaluation of the impact of environmental pollution on the health of the population, and exploration of a health risk assessment system. Therefore, we considered communities in the Wuhan metropolitan area as the research scale, combined residents’ spatial and temporal behaviors, and conducted a study on air pollution exposure patterns and environmental equity in Wuhan, in an attempt to accurately identify areas and populations at high risk of air pollution exposure and provide theoretical basis and policy recommendations for air pollution management in Wuhan.

## 3. Research Data and Methods

### 3.1. Study Area

Wuhan is a central city in central China, with a total area of 8569.15 km^2^ and a resident population of 11,212,000. Wuhan city is an important industrial base in China, and PM_2.5_ pollution in the city has been at a high level for a long time due to industrialization. Relevant studies have pointed out that PM_2.5_ pollution in Wuhan city has a serious impact on the health of its residents [36]. In this study, the metropolitan area of Wuhan, with a land area of 2820 km^2^ (Figure 1), was selected to help carry out regional management and better correspond to the characteristics of regional diffusion of pollutants [37]. Urban blocks were used as the basic unit for analyzing the spatial pattern of PM_2.5_ pollution exposure to represent the differences in pollution exposure between different urban spaces in Wuhan. Communities were used as the basic unit for analyzing the population differentiation of PM_2.5_ pollution exposure risk in Wuhan to reflect the differences in pollution exposure between different types of populations.

### 3.2. Study Data

The data sources used in this study were extensive and contained three main dimensions: air quality, built environment, and socioeconomic data. For air quality data, PM_2.5_ concentration monitoring data were obtained from the China Air Quality Online Testing and Analysis Platform (https://www.aqistudy.cn, accessed on 12 May 2020). The aerosol optical thickness data (AOD) were obtained from the Earth Observation Research Center for Data (EORC) and downloaded from the aerosol observation data of the Japan Aerospace Agency Himawari-8 (https://www.eorc.jaxa.jp, accessed on 12 May 2020).

Among the socioeconomic data, the demographic data consisted of the current population data of Wuhan city in 2017, the third statistical economic unit data of Wuhan city, the original data of social security individual residence and employment, and the original data of the trade union members’ residence and employment [38]. Smartphone LBS data were collected from Jiguang, a big data platform company, and the data included individuals’ locations within Wuhan from Monday, 17 April, to Sunday, 23 April, 2017. This is a full week with five working days and two rest days, with no special holidays, so it is universal and representative.

For the built environment data, the land data was Wuhan city 2017 land use data, the vegetation index data was NDVI data, and the road traffic data was from OpenStreetMap.

### 3.3. Study Methods

#### 3.3.1. Measurement of PM_2.5_ Concentration

To accurately analyze PM_2.5_ pollution exposure patterns and exposure risk equity in Wuhan, a geographically weighted regression model, and remote sensing inversion techniques were used to measure hourly PM_2.5_ concentrations and monthly average PM_2.5_ concentrations in Wuhan metropolitan development area, respectively. Firstly, PM_2.5_ concentration-related variables were selected, and then a geographically weighted regression model with the monitored PM_2.5_ concentration as the dependent variable and PM_2.5_ concentration-related variables as the independent variables was constructed to explore the relationship between the dependent and independent variables and derive the regression equation. Subsequently, a 1000-m-scale fishing net was constructed in GIS, and the geographic correlation variables of PM_2.5_ concentration at the plots of the fishing net were extracted and substituted into the regression equation to measure the PM_2.5_ concentration at the plots of the fishing net. On this basis, the kriging interpolation method was used to simulate the PM_2.5_ concentration within the Wuhan metropolitan development area, and the simulated PM_2.5_ concentration values of the study area were obtained. Finally, the average PM_2.5_ concentration within the city block was calculated as the PM_2.5_ concentration in the city block using the city block as the statistical unit. Wuhan city has 10 national monitoring stations, which are evenly distributed in the built-up area of Wuhan. They can theoretically cover an area of 25 square kilometers.

To measure the average PM_2.5_ concentration for each hour of a complete week in the Wuhan metropolitan development area, a geographically weighted regression model was constructed in this study, using data averaged for a year and calculated as follows:(1)$$\begin{array}{c} PM2.5_{it}=\beta_{0,it}+\beta_{1,it}AOD_{it}+\beta_{2,it}POP_{it}\\ +\beta_{3,it}NDVI_{it5}+\beta_{4,it}ROAD_{it}+\beta_{5,it}NP_{it}+\varepsilon_{it}, \end{array}$$
where $PM2.5{\;}_{it}$ (unit: μg/m^3^) is the average hourly PM_2.5_ concentration monitored at the monitoring station *i* at time *t*; $\beta_{0,it}$ denotes the location-specific intercept at time *t*; $\beta_{1,it}$-$\beta_{5,it}$ denotes the location-specific correlation variable coefficient at time *t*; $AOD_{it}$ (unitless) is the Japan Sunflower Satellite 8 AOD data at time *t*; $POP_{it}$ is the population density of the traffic cell where monitoring station *i* is located; $NDVI_{it}$ (unitless) is the MODIS NDVI value at monitoring station *i*; $NP_{it\;}$ is the Euclidean distance of monitoring station *i* from the major industrial pollution sources in Wuhan; and $\varepsilon_{it}$ is the error term of monitoring station i.

#### 3.3.2. PM_2.5_ Measurement under the Spatial and Temporal Activities of the Residents

(1)Measurement of PM_2.5_ pollution exposure in urban space based on residents’ spatial and temporal activities.

Air pollution exposure characterizes the exposure of a population to air pollution over a period of time, and therefore has a more direct link to public health. Identifying areas of high exposure to air pollution plays an important role in reducing the impact of air pollution on the health of the population. In this study, PM_2.5_ pollution exposure intensity was introduced as an indicator to study air pollution exposure. The population exposure risk assessment model proposed by Kousa et al. [39] was used to calculate the PM_2.5_ pollution exposure intensity values in urban blocks within the Wuhan metropolitan development area, which was based on the degree of population exposure to PM_2.5_ concentrations per unit area. Since the cell phone LBS data were collected from 17 March 2017 to 24 March 2017, a complete week of the year containing five working days and two holidays, the population density data for that week could be used instead of the annual data. The calculation formula is as follows:(2)$$E_{j}={\;\mathrm{NT}}_{ij}{\;*\;P}_{ij}{\;*A}_{ij}\;,$$
where $E_{ij}$ is the pollution exposure intensity of urban block *j* in μg − person/(m^3^ − km^2^) during a day, $A_{ij}$ is the population density of urban block *j* at time *i* (in person/km^2^), $P_{ij}$ is the air pollution concentration of urban block *j* at time *i*, ${\mathrm{NT}}_{ij}$ is the cumulative time of air pollution in urban neighborhoods *j* (in hours).

(2)PM_2.5_ pollution exposure measurement of urban population based on residents’ spatial and temporal activities.

In this study, we set the time for daily travel activities for residents of different age groups. Children and the elderly were designated as vulnerable groups because of health problems and inability to drive motor vehicles due to mobility problems; these groups prefer to spend their time within proximity [15]. Therefore, we assumed that both the elderly (65+ years) and children (0–6 years) spend the day in their residential community. In contrast, the working population (19–59 years) was at their place of work between 8 a.m. and 6 p.m. on Monday through Friday and at home at all other times, that is, the working population spent 28.58% of their time in the workplace. The calculation formula is:(3)$$POD_{ij}=\frac{\sum [P_{i}\left(COt_{ij}+CDt_{ij}\right)]}{\sum P_{i}*24},$$

$POD_{ij}$ is the weighted PM_2.5_ concentration value in the urban community *i* during a day, $P_{i}$ is the number of people in the same group of OD relationships in urban community *i*, ∑$COt_{ij}$ is the sum of hourly PM_2.5_ concentration exposure outside of working hours for the population residing in the urban community *i* in month *j*, ∑$CDt_{ij}\;$ is the sum of working hours for the population residing in the urban community *i* at the work location in month *j*, PM_2.5_ concentration exposure in j urban community *i*, and ∑$P_{i}$ is the sum of all the total population living in an urban community *i*.

#### 3.3.3. Walk Score

A walk score is a representative and internationally accepted method for measuring walkability [40,41]. The process of calculating the walkability index of urban blocks was as follows: first, the weights of different types of services and facilities were determined, and the basic walkability index was obtained by decaying according to the law of distance decay; then, the influence of intersection density and block scale was superimposed, and the walkability index was modified to obtain the single-point walkability index. Finally, the single-point walkability index was obtained using the spatial statistics method with the statistical unit of urban blocks and the population weight. Finally, the spatial statistical method was used to weight the single-point walking index by using the population weight as the statistical unit of the city block to obtain the surface area walking index of the city block (Figure 2).

The specific steps of the walking index measurement are as follows.

(1)Measurement of basic walking index.

The basic walking index is obtained by calculating the distance from a certain point to different service facilities within a certain range. The higher the number of service facilities, the richer the category, and the closer the distance, the higher the calculated basic walking index. The different service facilities are weighted according to their importance and substitutability, as shown in the following table (Table 1).

On this basis, the urban development area of Wuhan is divided into 2811 square fishnet grids with a side length of 1000 m as the calculation unit. Using the center of mass point of the fishing net as the starting point, the basic walking index of the starting point is calculated by considering the acceptable walking time and distance for pedestrians and attenuating the weights of different services and facilities. When the distance is within 500 m, no attenuation occurs; between 500 m and 1000 m, the attenuation rate is 25%; between 1000 m and 1500 m, the attenuation rate is 88%; when the service facilities are more than 1500 m away from the starting point, the attenuation rate is 1. After calculating the distance attenuation, the weights of various service facilities around the starting point are summed up to obtain the basic walking index of the starting point.

(2)Measurement of the single-point walking index.

After obtaining the base walking index of the departure point, the single-point walking index was obtained by further decaying the base walking index according to the intersection density and block scale within the grid cell. Among them, the intersection density and block scale decay rates are divided into five levels each, and the maximum decay rate is 10% (Table 2).

(3)Measurement of the surface area walking index.

After calculating the single-point walking index, the face area walking index of urban neighborhoods was calculated using the population weight of the grid as the weight. The population weight of the grid is the ratio of the number of people in the grid to the total population, and the number of people in the grid is obtained by spatially connecting the grid mass to the Wuhan city community demographic data.

## 4. Results

### 4.1. Spatial Pattern of PM_2.5_ Pollution Exposure

First, the hourly PM_2.5_ concentrations in Wuhan were measured, and the calculation results showed that the R^2^ were all above 0.6, indicating that the model fit was good. The daily average of 9:00 a.m., 12:00 p.m., and 18:00 p.m. were selected as the displayed results (Figure 3). Overall, PM_2.5_ concentrations show a spatial pattern of high outside and low inside, and high west and low east. Urban blocks with high PM_2.5_ concentrations are mainly the recipients of high pollution enterprises in the “industrial upgrading” of cities. The PM_2.5_ concentrations within the Third Ring are in a polycentric pattern, the PM_2.5_ within the first ring, second ring, and third ring differed a little, and the median and mean were all between 33 ± 0.1, with no statistically significant differences. Higher concentrations concentrated in the Hankou area within the First Ring, the Wuhan Iron and Steel plant area in the Qingshan District and the Optics Valley area. In addition, the PM_2.5_ concentrations in large ecological spaces such as near East Lake are relatively low.

The PM_2.5_ pollution exposure intensity of each urban block in Wuhan was calculated based on the spatial and temporal activities of residents. According to the hourly PM_2.5_ pollution exposure intensity of urban neighborhoods, the quantile classification method was used to classify urban neighborhoods into high, medium, and low levels, and were assigned values of 3, 2, and 1, respectively, for a total of 168 classification assignments. On this basis, the hourly quantile values of urban neighborhoods were summarized, and the quantile classification method was used to classify the urban blocks into high, medium, and low levels. At the same time, the natural interruption point grading method was used to classify urban blocks into high, medium, and low levels according to the size of the walkability index. The results of the two categories of classification were combined, and seven categories of urban blocks with PM_2.5_ pollution exposure risk were identified. The spatial pattern of PM_2.5_ pollution exposure in Wuhan city is shown in Figure 4.

According to the classification map of PM_2.5_ exposure risk in urban blocks in Wuhan, the “high-high” urban blocks were mainly located in the core area of urban functions where the Yangtze and Han rivers meet within the second ring and were adjacent to the main roads, showing the characteristics of concentration within the first ring and expansion along the second ring.

High-medium and medium-high urban blocks were distributed contiguously within the Third Ring Road and dotted outside the Third Ring Road, and the exposure risk was high on the inside and low on the outside. These two types of urban blocks are mostly used as service centers of the district, with a mature built environment, high walkability, and dense population activities; however, because of the limitations of the economic development stage, the protection of the environment is neglected, resulting in more serious air pollution.

Low-high class urban blocks were mainly distributed in urbanized areas adjacent to large natural ecological spaces within the third ring, and low-medium class urban blocks were distributed in semi-urbanized suburban areas outside the third ring, adjacent to medium-high class urban blocks internally and surrounded by low-low class urban blocks externally, with low exposure risk. Large-scale ecological spaces can significantly improve the air environment, while the proximity to the central area makes such areas nearby more walkable and densely populated, which plays a positive role in health [13].

### 4.2. PM_2.5_ Pollution Exposure for Different Age Groups

This study combined the measured PM_2.5_ concentrations and the spatial distribution of population residence and workplaces based on LBS to measure the PM_2.5_ pollution exposure risk of community populations under different spatial and temporal activities (Figure 5). There were differences in the weighted PM_2.5_ concentrations among communities, with an overall spatial pattern of “high in the northwest and low in the southeast, with finger-like dispersion”, which may be due to high population density, industrial clustering, and high automobile use in these urban areas. By comparing the PM_2.5_ concentration and the weighted PM_2.5_ concentration, the average weighted PM_2.5_ concentration was found to be 88.34 µg/m^3^, which was slightly higher than the average PM_2.5_ concentration of 88.28 µg/m^3^. This suggested that in Wuhan city, ignoring the spatio-temporal behavior of residents and using average concentrations to represent PM_2.5_ pollution intensity underestimated the actual exposure hazard of PM_2.5_ pollution to residents.

Based on the proportion of the elderly (65+), children (0–6), and working population (19–59) in each community, the communities were classified into five classes: high, medium-high, medium, medium-low, and low, using the natural interruption point grading method, to obtain the distribution maps of high, low, and working age population (Figure 6). In terms of spatial distribution, the elderly people communities were mainly distributed between the first and third rings; the junior communities were distributed in the suburbs of the city near the third ring road, and the working-age communities were distributed in the middle of the second to third rings and along the third ring road.

Based on the weighted PM_2.5_ concentration and the proportion of the population of different age structures in each community, the communities were classified into high, medium, and low levels using the natural interruption point grading method. By combining the classification results of the two indicators, nine categories of PM_2.5_ pollution exposure risk communities could be classified from high PM_2.5_ concentration–high proportion of elderly to low PM_2.5_ concentration–low proportion of elderly.

Communities were divided according to population age attributes, and all communities were divided into equal deciles according to age attributes and arranged in ascending order. High-population decile communities had a larger proportion of the population in that age group. The weighted PM_2.5_ concentrations were calculated separately for each decile community, and the age-specific differences in PM_2.5_ pollution exposure risk were compared (Figure 7).

(1)The exposure risk of the elderly (65 years old and above) was higher than that of other age groups.

As shown in Figure 8, the weighted PM_2.5_ concentration increased steadily with the increase in the proportion of elderly people, and the weighted PM_2.5_ concentration in the community with the highest proportion of elderly people was 88.77 mg/m^3^ (D10), whereas the weighted PM_2.5_ concentration in the community with a higher number of residents of other age groups was 88.01 mg/m^3^ (D1), which indicates that the elderly people had a higher risk of PM_2.5_ pollution exposure than other age groups.

(2)Communities with high exposure risk for the elderly were concentrated around the center and sub-center of the city.

By comparing the spatial distribution of PM_2.5_ concentration for the elderly (65 years old and above) and the population-weighted PM_2.5_ concentration (Figure 8), communities with high exposure risk for the elderly were found to be concentrated around the main center and sub-center of the city. High-weighted PM_2.5_ concentration–high number of elderly communities were mainly distributed between the first and third rings. The high-risk elderly were mainly distributed around the main urban center and sub-center of the city. The high density of production and living and high volume of motor vehicle traffic in each level center is likely to cause air pollution problems and adversely affect the health of the elderly.

(3)The exposure risk of children (0–6 years old) was not significantly different from that of the other age groups.

As shown in Figure 7, the weighted PM_2.5_ concentration showed a normal distribution with the increase of the proportion of children, that is, it first increased and then decreased and did not differ significantly from those of the other age groups.

(4)Communities with high exposure and high risk of children were scattered in the suburbs of the city, showing a high outer and low inner distribution.

By comparing the spatial distribution of children (0–6 years old) and weighted PM_2.5_ concentration (Figure 8), we found that high weighted PM_2.5_ concentration–high number of children communities were mainly distributed in the suburban areas outside the third ring road. The suburban areas have certain job opportunities and lower housing prices, where children live together with the elderly and parents, but the higher PM_2.5_ concentrations and fewer ecological green areas in the suburban areas exposed these groups to higher PM_2.5_ pollution exposure risks.

(5)The risk of PM_2.5_ pollution exposure was relatively low in the working-age population (19–59 years).

Figure 7 shows that the weighted PM_2.5_ concentration decreased gradually with an increase in the proportion of working-age people, with the highest proportion of working-age people having a weighted PM_2.5_ concentration of 87.81 mg/m^3^ (D10) and the communities with more residents of other age groups having a weighted PM_2.5_ concentration of 88.19 mg/m^3^ (D1). These results suggest that the risk of PM_2.5_ exposure was lower for working-age populations than for the other age groups and that this phenomenon was more pronounced as the proportion of working-age populations increased, possibly because communities with high working-age populations were located near large natural ecological spaces where industries were less prevalent and air quality was relatively good. These findings further suggest that vulnerable groups, such as the elderly, were at higher risk of PM_2.5_ pollution exposure.

(6)Cluster distribution of communities with high exposure risk for working-age people outside the first ring.

By comparing the spatial distribution of the working-age population (19–59 years) and weighted PM_2.5_ concentration, the communities with high exposure risk for the working-age population were found to be distributed in clusters outside the first ring road. These areas were mainly the recipients of out-migrating industries, with lower housing prices and more job opportunities, but also faced higher PM_2.5_ pollution exposure risks.

### 4.3. Health Equity within Vulnerable Populations

(1)Among vulnerable groups, a few bore higher exposure risks.

To further investigate the equity of the exposure risk for vulnerable people, the Gini coefficients of community-weighted PM_2.5_ concentration and the number of vulnerable people were calculated using Python and then plotted as Lorenz curves. The Gini coefficient was 0.48 for the weighted PM_2.5_ concentration and the number of elderly people, and 0.58 for the weighted PM_2.5_ concentration and the number of children, both indicating a “large gap”.

Based on the Lorenz curves (Figure 9), 40% of children were exposed to 80% of the weighted PM_2.5_ concentration, and 40% of the elderly were exposed to more than 70% of the weighted PM_2.5_ concentration, with a large exposure risk gap and significant health inequity.

(2)Spatial heterogeneity in areas with high exposure risk.

GeoDa software was used to conduct bivariate LISA analysis of weighted PM_2.5_ concentration and number of elderly people, and weighted PM_2.5_ concentration and number of children, respectively, to further analyze the equity of their spatial distribution. The results showed that the Moran index of weighted PM_2.5_ concentration and the number of elderly people was 0.179, with a *p*-value of 0.001 and a z-score of 15.9054, indicating that the clustering of weighted PM_2.5_ concentration and elderly people was extremely high, that is, the communities with high weighted PM_2.5_ concentration and the main gathering activities of elderly people were continuously spatially distributed (Figure 10). The Moran index of the weighted PM_2.5_ concentration and the number of children was 0.179, with a *p*-value of –0.023 and a z-score of –2.02, indicating that the weighted PM_2.5_ concentration and the distribution of children were significantly discrete, that is, communities with high weighted PM_2.5_ concentrations where children mainly gathered for activities were spatially dispersed (Figure 10).

LISA analysis was performed for PM_2.5_ concentration and the number of elderly people, PM_2.5_ concentration and the number of children and four types of communities with significant local variability were obtained (Figure 11).

(3)High internal and low external exposure risk for the elderly and high internal and low external exposure risk for children.

According to the clustering results of LISA analysis of weighted PM_2.5_ concentration and the number of vulnerable people, the risk of PM_2.5_ pollution exposure in the elderly showed a spatial differentiation of activities with high internal and low external exposure. Among 1819 communities in the metropolitan area of Wuhan, 163 had a high weighted PM_2.5_ concentration–high number of elderly people; these communities were concentrated between the first and third rings. Among these, the Wuchang and Jiang’an districts were concentrated and distributed in a large number. There were 112 low-weighted PM_2.5_ concentration–high elderly population communities located near the ecological environment along the third ring road. This was mainly because elderly people tend to gather in and around the inner-city center of the third ring road, where the high-density production, living, and high volume of automobile traffic are likely to cause air pollution problems, thus threatening the elderly population in the area. The lack of ecological green space further exacerbates this risk. On the other hand, the suburban area attracts the elderly population due to the convenience brought by the proximity to the central area. There are large-scale ecological spaces such as East Lake, Moon Lake, and South Lake, which have significant effects on reducing air pollution and play a positive role in the health of the elderly. There were 120 communities with high weighted PM_2.5_ concentration–low elderly population and 268 communities with low weighted PM_2.5_ concentration–low elderly population, all of which were located in the periphery of metropolitan areas, and the risk of PM_2.5_ pollution exposure in these spaces was relatively low due to the sparse population distribution. Overall, the PM_2.5_ pollution exposure risk of elderly people showed a spatial variation of high on the inside and low on the outside, reflecting the unevenness of spatial risk, that is, the existence of inequity.

In contrast to the elderly, children’s PM_2.5_ pollution exposure risk showed a spatial differentiation of activities with low internal and high external exposure. There were 49 communities with high-weighted PM_2.5_ concentration–high number of children among 1819 communities in the metropolitan area of Wuhan and that were discrete in the suburban areas along the third ring road. These areas, as catchment areas for heavy industries, offer many employment opportunities and have low-cost housing that attracts young people, most of whom are of childbearing age, thus increasing the number of children in the area. The 141 communities with low weighted PM_2.5_ concentration–high number of children were located near the ecological space outside the third ring road, and these areas were also the employment centers of the district, where the spatial differentiation of the young generation lived as high housing prices forced them to work nearby and live together with their children. There were 131 communities with high-weighted PM_2.5_ concentration–low number of children and 228 communities with low-weighted PM_2.5_ concentration–low number of children, all of which were located in the periphery of the metropolitan area, and these areas were at low risk. Overall, the risk of children’s PM_2.5_ pollution exposure showed a spatial differentiation of activities with low internal and high external exposure, and the presence of inequity.

## 5. Discussion

In general, urban centers are built with a mature environment and are rich in services and pedestrian systems, which attracts people to gather and move, thus generating a large amount of motorized traffic and increasing the PM_2.5_ concentration. Under the law of land rent, large ecological spaces are mostly located at the periphery of the city, and the urban blocks around them enjoy the convenience of the urban environment as well as clean air brought by the natural environment. Thus, the spatial pattern of PM_2.5_ exposure risk in urban blocks shows the circle differentiation of “high on the inside and low on the outside, decreasing circles” and the distribution characteristics of “central concentration, axis extension” in high-risk urban blocks.

Vulnerable groups such as the elderly and children are disproportionately exposed to PM_2.5_ pollution and their spatial distribution varies significantly, with the risk of PM_2.5_ pollution exposure being higher on the inside and lower on the outside for the elderly and higher outside and lower inside for children. The main reason for this phenomenon is the urban governance corporatism and the capital-led urbanization model that accompany the urbanization process in China. Capital-led urban space production neglects the survival and development of human beings, resulting in the alienation of space and eventually the over-capitalization of public space. The high-quality, healthy spaces in the city are inevitably developed for commercial purposes, increasing the economic value of the land. Therefore, industrial spaces and reserved ecological spaces are mostly located in the low-priced areas in the suburbs of the city. The aging phenomenon in China is becoming more pronounced, with older people living mostly in older neighborhoods in urban centers, enjoying the benefits of a built environment but also suffering from the threat of air pollution due to widespread and massive motor vehicle emissions and a lack of ecological space. China’s fertility rate has been declining in recent years, and despite the government’s “three-child policy,” the willingness of young people to marry and have children remains low due to China’s rising housing prices. With the recent urban sprawl and relocation of downtown industries, many large corporations have moved to the suburbs, and young people tend to move there as well, as jobs are available and housing costs are lower, and then start a family there until they are middle-aged and purchase a second, improved home; therefore, their children are often exposed to severe industrial air pollution during their childhood.

To mitigate urban PM_2.5_ air pollution and address the injustice of air pollution exposure to the elderly and children, the following urban planning policies and recommendations are proposed:(1)Neighborhoods at high risk of PM_2.5_ pollution among the elderly are mainly located in high-intensity built-up areas in the city center, for which, in conjunction with China’s ongoing urban renewal initiatives, this study proposes a morphology-guided, functionally mixed strategy. We propose that the guidance of block texture and morphology should be used, an urban ventilation system should be constructed, the internal ventilation corridor of the block should be built, and the internal ventilation corridor should be linked with the urban ventilation corridor to form a “sewage system”. Improving the mix of functions can reduce the travel distance and the probability of motor vehicle travel for urban residents, and reduce the total carbon emission and path exposure of residents’ travel;(2)The common feature of typical high-risk areas for PM_2.5_ pollution exposure for the elderly and children is the insufficient construction of green and open spaces. The study proposes the strategy of increasing their extent, improving planning, and optimizing the layout. A scattered layout should be avoided in the process of configuring green space, and the centralization and scale of green space should be promoted. Regarding the layout of green and open space, the layout should be optimized according to the functional characteristics;(3)As a road connecting urban areas, the trunk road is characterized by high traffic volume and high pollution. In the dense area of the trunk road network, the study proposes the protection retreat line restriction function strategy. During the construction process, the threshold distance of significant reduction of pollutants on both sides of the road should be calculated as the basis of the protection retreat line. Strict functional construction restrictions should be made in the high concentration range to avoid crowd-gathering spaces such as schools, residences, and open spaces from being too close to the main roads to reduce the risk of PM_2.5_ pollution exposure;(4)During the construction of new cities, emphasis should be placed on the integration of industries and cities, and the location of large industrial enterprises should not only focus on industrial benefits but should also consider ecological benefits. The residential land of large industrial enterprises should be close to large ecological spaces to reduce industrial waste gas pollution to the residents. At the same time, adequate protective measures should be set up outside the industrial land of concentrated scale to guide the pollutants to stop their diffusion to the isolation and absorption green belt and avoid their flow to the crowd-gathering space.

In this study, five types of PM_2.5_ concentration built environment influencing factors, namely AOD (aerosol data), road traffic, population density, land use, and industrial pollution impact, were integrated to simulate high precision hourly PM_2.5_ concentrations and measure PM_2.5_ pollution exposure intensity in urban neighborhoods by combining the dynamic population distribution characterized by hourly precision LBS data. On this basis, the spatial pattern of PM_2.5_ pollution exposure risk is further explored in conjunction with the built environment walkability, revealing the actual PM_2.5_ pollution exposure risk in the built environment and enriching the understanding of the built environment air pollution exposure risk pattern based on the dynamic distribution of population and PM_2.5_. This study provides a feasible method and scientific basis for studying the differences in air pollution exposure among different groups and the derived environmental justice issues, and for accurately identifying the high-risk areas and populations of air pollution exposure in urban areas.

This study has a few shortcomings. (1) Due to data limitations, the time period studied in this research is one week, and although that week is a full week with five working days and two rest days, it is still contingent in the calculation in the measurement of PM_2.5_. In future studies, the use of data with longer time spans could be considered for the study. (2) In the study of community population differentiation of PM_2.5_ pollution exposure, the pathway exposure generated by different groups of traffic travel processes was not calculated, which could lead to some deviation between the calculated results of PM_2.5_ pollution exposure and the actual exposure of the population. Future studies should divide the population according to their daily behavioral patterns, simulate their traffic travel paths according to their behavioral characteristics, calculate their traffic travel exposure levels, and more comprehensively assess the differences in PM_2.5_ pollution exposure among different populations. (3) There are currently four main approaches to air pollution exposure measurement, including proximity models, atmospheric dispersion models, interpolation models, and regression models [42,43]. In general, the proximity model is a qualitative method to determine whether space or individuals are exposed to the effects of pollution sources based on distance, and is not applicable to the study of air pollution exposure on a region-wide scale. The atmospheric dispersion model is built under different assumptions and can only be applied to a certain range, and is not applicable to the study of regions with large environmental differences. In comparison, the regression model and interpolation method take into account the variability of geographic environments, and have higher accuracy and applicability. In this study, we measured the PM_2.5_ distribution in Wuhan City using the technique with an accuracy of 1 KM×1 KM. The accuracy and fit are high enough to support the conclusion. However, the multi-variate models and kriging techniques may not necessarily capture the complex urban PM_2.5_ distribution. Nowadays, various intelligent methods such as neural network algorithms and their improvement algorithms have emerged and have initially been applied to the measurement of urban PM_2.5_ and PM10 pollutant concentrations. For example, Perez et al. [44] predicted urban PM_2.5_ hourly concentration using multilayer neural network and linear regression method, Grivas et al. [45] combined a genetic algorithm and neural network method to predict PM10 hourly concentration, Zou Bin et al. [46] measured urban PM_2.5_ based on BP artificial neural network, and all of them concluded that the neural network method is better than the linear regression method and kriging interpolation method. In the following research, we will improve the method.

## 6. Conclusions

This study systematically analyzes the exposure risk of the spatial pattern of urban PM_2.5_ pollution and its variability based on the dynamic distribution of population activities and PM_2.5_ concentrations, as well as the health equity of PM_2.5_ pollution exposure risk based on the spatial and temporal activities of the residents. Compared with the traditional static PM_2.5_ pollution exposure assessment, the dynamic assessment method also considers the urban population and the high mobility of air pollution, thus accurately revealing the actual risk of air pollution. This study enriches our understanding of PM_2.5_ pollution exposure risk patterns in the built environment, expands the understanding of population differentiation of PM_2.5_ pollution exposure risk for the spatial and temporal activities of residents, and provides effective references and suggestions for urban planning decision makers and program designers to improve air quality, reduce PM_2.5_ pollution exposure risk, and promote residents’ health equality in a targeted manner. [47]

## Figures and Tables

**Figure 1 ijerph-19-12671-f001:**
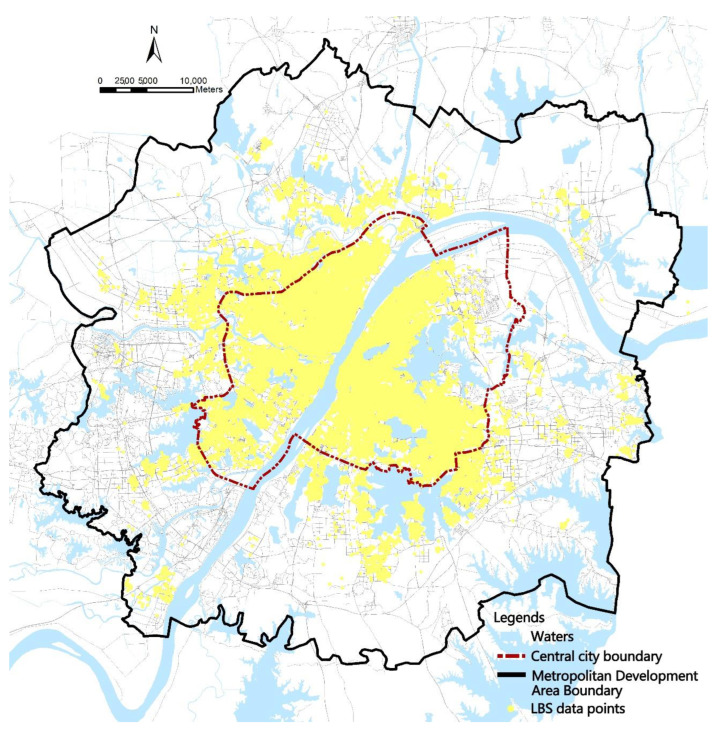
Wuhan city district map.

**Figure 2 ijerph-19-12671-f002:**
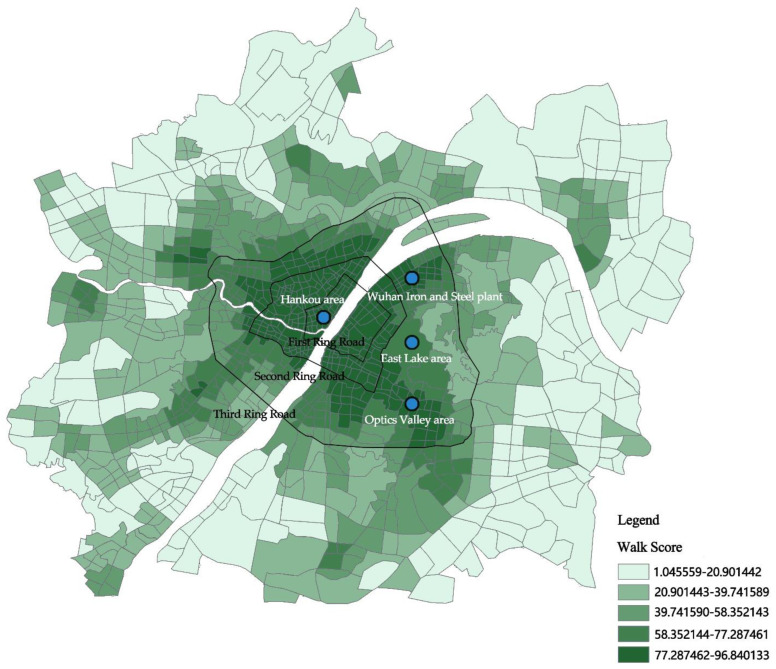
Wuhan metropolitan area walking index score.

**Figure 3 ijerph-19-12671-f003:**
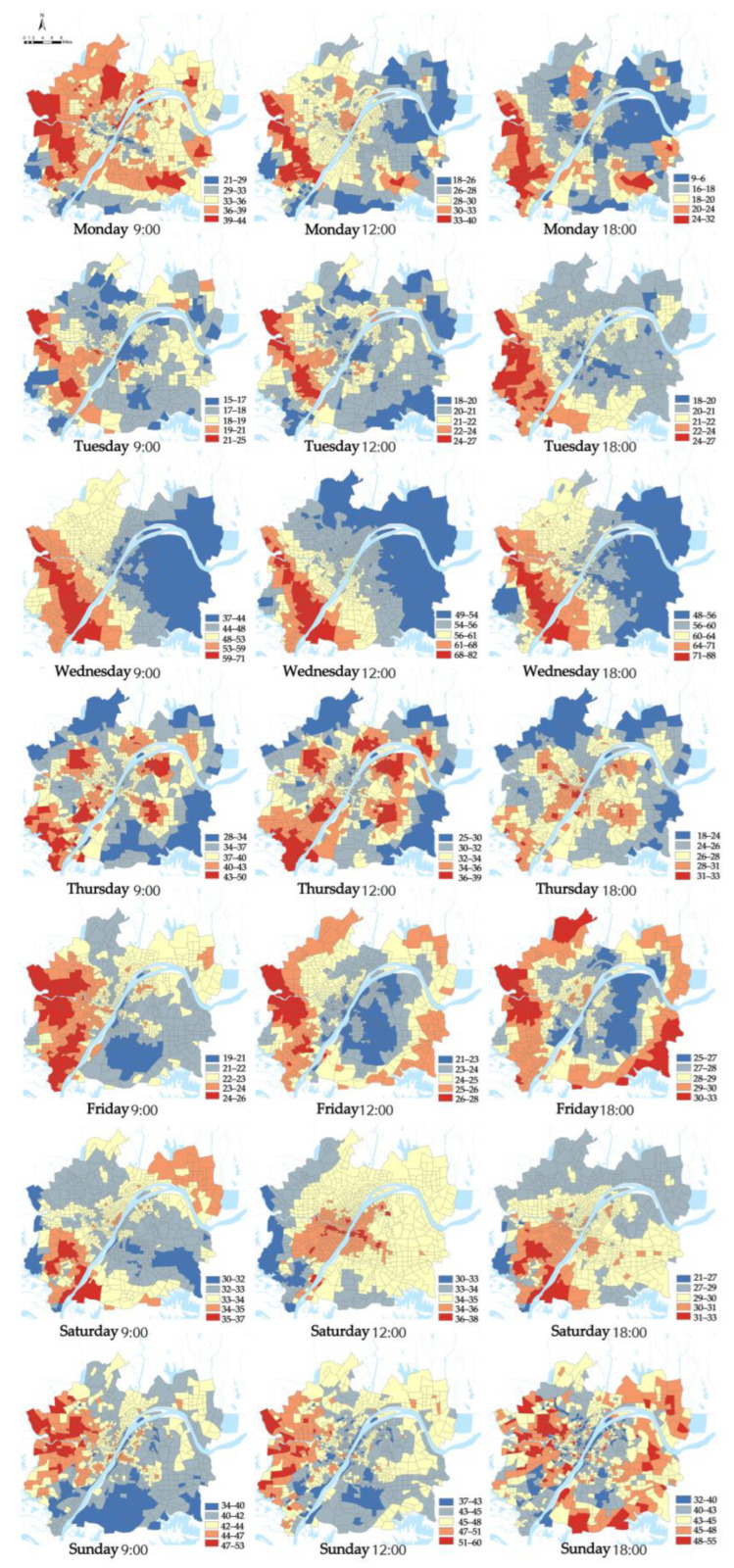
Spatial distribution of hourly PM_2.5_ concentrations in the urban blocks of Wuhan.

**Figure 4 ijerph-19-12671-f004:**
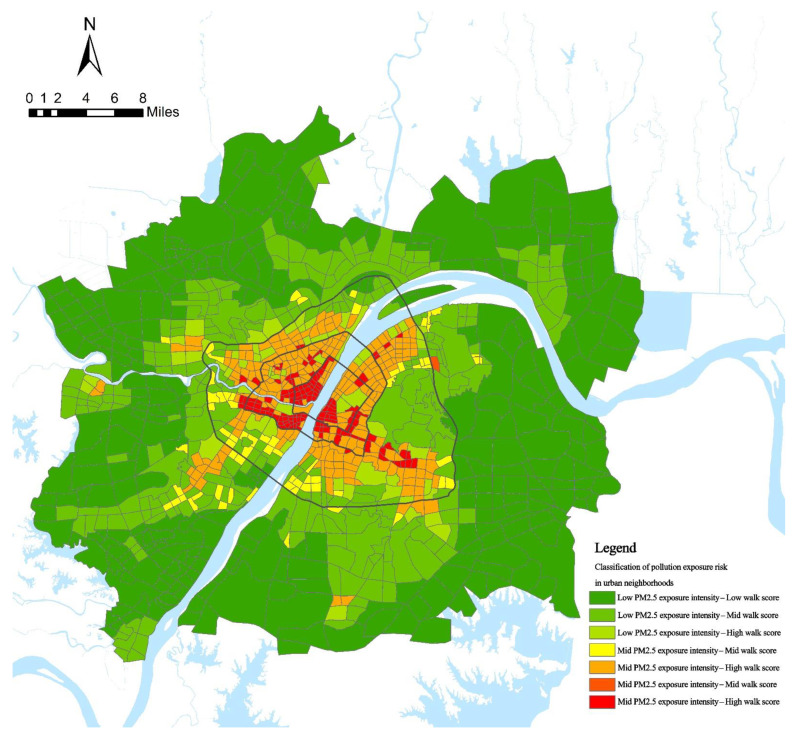
Risk classification of PM_2.5_ pollution exposure in urban blocks.

**Figure 5 ijerph-19-12671-f005:**
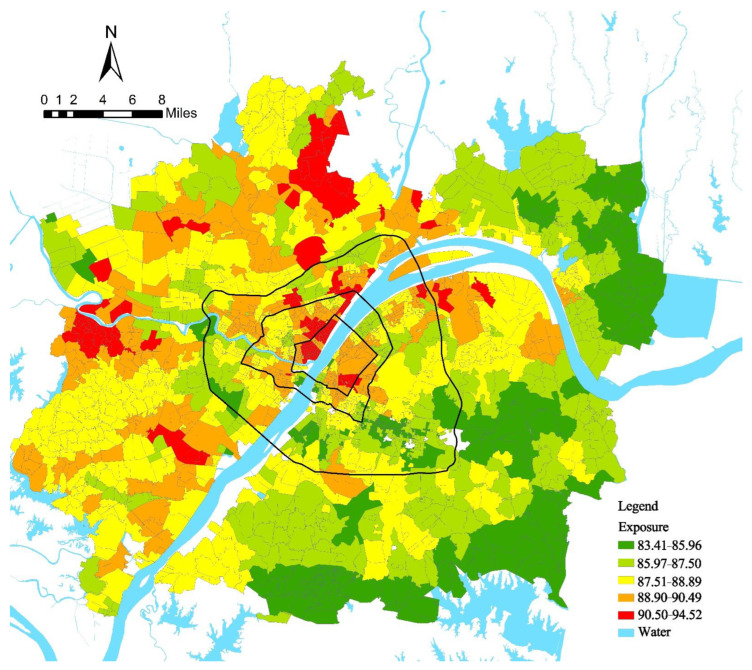
Spatial distribution of weighted PM_2.5_ concentration.

**Figure 6 ijerph-19-12671-f006:**
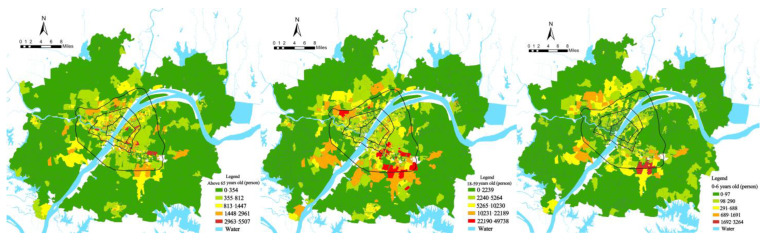
Distribution of elderly people community, working age community, and children community.

**Figure 7 ijerph-19-12671-f007:**
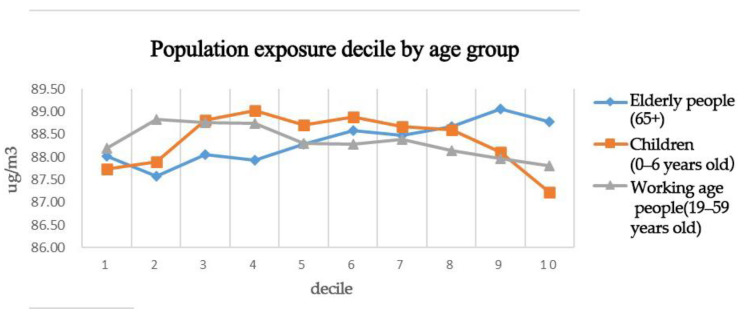
Population decile-weighted PM_2.5_ concentrations by age group.

**Figure 8 ijerph-19-12671-f008:**
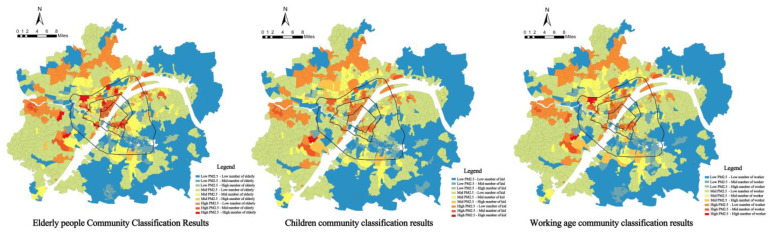
Community PM_2.5_ pollution exposure risk classification results.

**Figure 9 ijerph-19-12671-f009:**
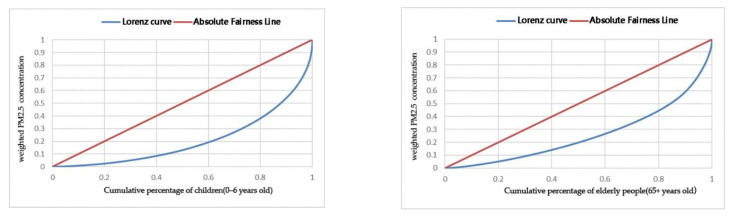
PM_2.5_-Lorenz curve for the elderly people and children.

**Figure 10 ijerph-19-12671-f010:**
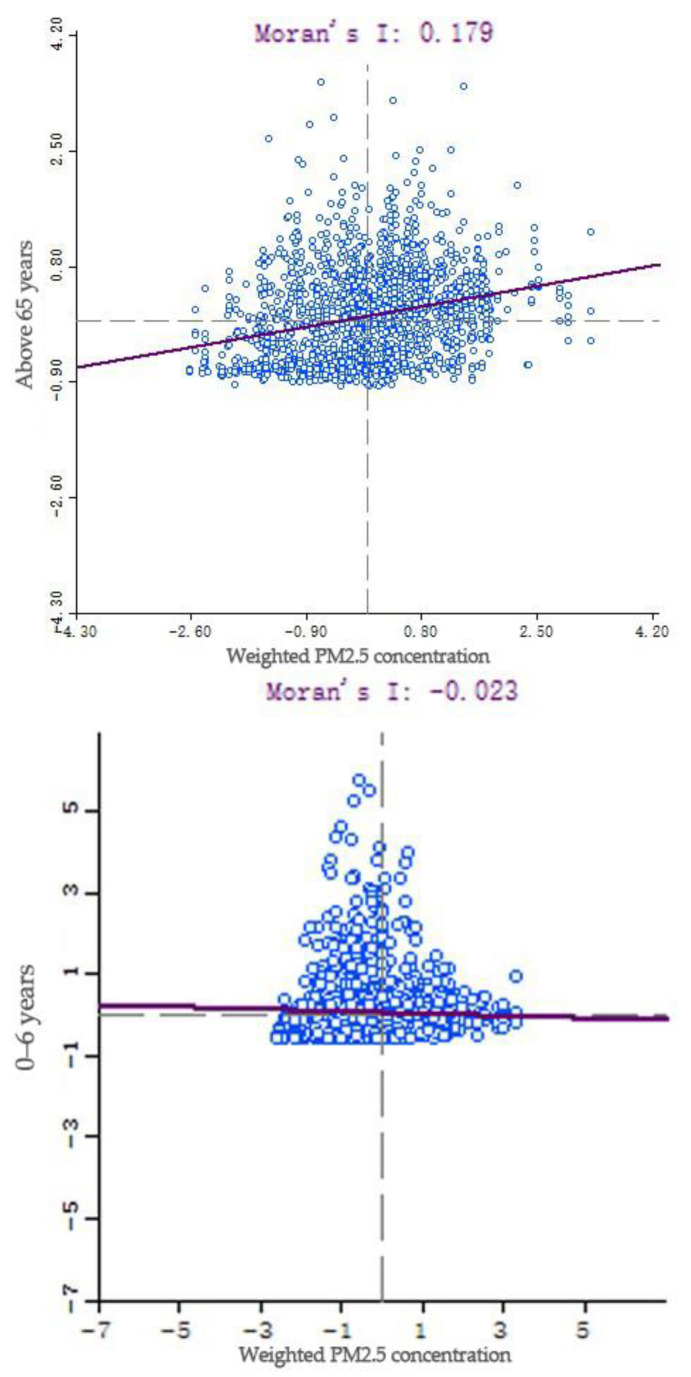
PM_2.5_–elderly people, children’s Moran index.

**Figure 11 ijerph-19-12671-f011:**
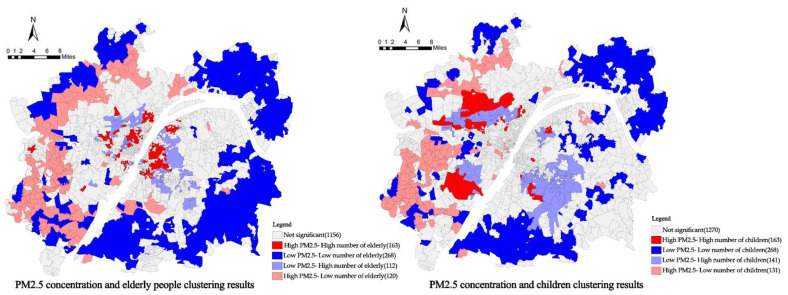
PM_2.5_ concentration and elderly people clustering results, PM_2.5_ concentration and children clustering results.

**Table 1 ijerph-19-12671-t001:** Classification and weighting of service facilities.

Facility Classification	Name of Facility	Weights
Shopping	Grocery	3
	Mall	2
Dining	Restaurants	3
	Café	2
Leisure	Bookstore	1
	Park	1
	Entertainment Venue	1
Public Services	Hospital	1
	School	1
	Bank	1
Total		16

**Table 2 ijerph-19-12671-t002:** Intersection density, block-scale attenuation table.

Block Length (m)	Decay Rate (%)	Intersection Density (Pcs/Square Mile)	Decay Rate (%)
<120	0	>200	0
120–150	1	150–200	1
150–165	2	120–150	2
165–180	3	90–120	3
180–195	4	60–90	4
>195	5	<60	5

## Data Availability

Not applicable.

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
