# Peer review of "Spatial Equity of PM2.5 Pollution Exposures in High-Density Metropolitan Areas Based on Remote Sensing, LBS and GIS Data: A Case Study in Wuhan, China"

_ijerph, 2022, doi:10.3390/ijerph191912671_

Round 1

Reviewer 1 Report (Previous Reviewer 3)

The authors took into account most of the comments, which improved the understanding of the text.

   However, some remarks remain:

1. The authors use different designations for some groups (in particular children/kids). In Fig.6,7,8, a single designation of age groups is better: children (0-6 years old), working age (19-59 years old), elderly people (65+).

2. In fig. 6 the fragment "19-59" is duplicated and the group "children" is missing

 3. The designation "high-low" is not clear (what refers to the population and what to the impact) in the legend (Fig. 11)

Author Response

The authors took into account most of the comments, which improved the understanding of the text.

Response:Thank you for your evaluation and encouragement. We greatly appreciate your insightful and constructive comments, which help to improve the quality of the manuscript greatly. Point-to-point responses to all the comments and suggestions are listed below.

However, some remarks remain:

  1. The authors use different designations for some groups (in particular children/ kids). In Fig.6,7,8, a single designation of age groups is better: children (0-6 years old), working age (19-59 years old), elderly people (65+).

Response1:Thank you for your suggestion. I have replaced kids with children and seniors with elderly people in all text and figures.

  1. In fig. 6 the fragment "19-59" is duplicated and the group "children" is missing

Response2:Thank you for your suggestion. I made the correct changes.

  1. The designation "high-low" is not clear (what refers to the population and what to the impact) in the legend (Fig. 11)

Response3:Thank you for your suggestion. I have added the complete information in the legend.

Reviewer 2 Report (New Reviewer)

The work is interesting a fit the journal. However, there are several major corrections:

a) English grammar is poor and you should summarize several sections;

b) Some plots show a very poor quality, please adjust;

c) The introduction lacks in many parts, such as low-cost sensors. As well as there is no comparison with other references. This is important for a study where only a certain type of measurement has been recorded.

Please add:

Baldelli, A., 2021. Evaluation of a low-cost multi-channel monitor for indoor air quality through a novel, low-cost, and reproducible platform. Measurement: Sensors17, p.100059.

Castell, N., Schneider, P., Grossberndt, S., Fredriksen, M.F., Sousa-Santos, G., Vogt, M. and Bartonova, A., 2018. Localized real-time information on outdoor air quality at kindergartens in Oslo, Norway using low-cost sensor nodes. Environmental research165, pp.410-419.

Roca, D., Lagüela, S., Díaz-Vilariño, L., Armesto, J. and Arias, P., 2013. Low-cost aerial unit for outdoor inspection of building façades. Automation in Construction36, pp.128-135.

Motlagh, N.H., Zaidan, M.A., Fung, P.L., Li, X., Matsumi, Y., Petäjä, T., Kulmala, M., Tarkoma, S. and Hussein, T., 2020, November. Low-cost air quality sensing process: Validation by indoor-outdoor measurements. In 2020 15th IEEE Conference on Industrial Electronics and Applications (ICIEA) (pp. 223-228). IEEE.

I would be happy to revise this paper again.

Round 2

Reviewer 2 Report (New Reviewer)

The work can be accepted as it is.

This manuscript is a resubmission of an earlier submission. The following is a list of the peer review reports and author responses from that submission.

Round 1

Reviewer 1 Report

Generally, the scientific contributions, innovative points and engineering application potential should be well summarized and improved. Some gaps still exist in the present version, specifically in Section 3 and 4, and further improvement and polishment are required for the revised manuscript. Currently, the method section is very vague, please provide some details and remove the useless information. In the manuscript, there are many word spelling errors and grammar errors. In addition, many sentences are very wordy and do not provide useful information. Overall, the english language is painful to read. Pls consider polish your language through your coauthors and native english speakers. 

Currently, the quantification of these indicators is very rough. Overall, further details of methods are needed, and an uncertainty analysis of methodological part needs to be conducted in the discussion section. It is not common to use project symbol to summarize results. You used many similar undefined terms, and they are not common. Please check all your manuscript.

 â€¯  

Author Response

Response to Reviewer 1 Comments

Generally, the scientific contributions, innovative points and engineering application potential should be well summarized and improved. Some gaps still exist in the present version, specifically in Section 3 and 4, and further improvement and polishment are required for the revised manuscript. Currently, the method section is very vague, please provide some details and remove the useless information. In the manuscript, there are many word spelling errors and grammar errors. In addition, many sentences are very wordy and do not provide useful information. Overall, the english language is painful to read. Pls consider polish your language through your coauthors and native english speakers. 

Currently, the quantification of these indicators is very rough. Overall, further details of methods are needed, and an uncertainty analysis of methodological part needs to be conducted in the discussion section. It is not common to use project symbol to summarize results. You used many similar undefined terms, and they are not common. Please check all your manuscript.

Response:Thank you for your evaluation and encouragement. We greatly appreciate your insightful and constructive comments, which help to improve the quality of the manuscript greatly. Point-to-point responses to all the comments and suggestions are listed below.

Point 1:Generally, the scientific contributions, innovative points and engineering application potential should be well summarized and improved.

Response1:Thank you for your suggestion. I have made the following changes:

“In this study, five types of PM2.5 concentration built environment influencing factors, namely AOD (aerosol data), road traffic, population density, land use and industrial pol-lution impact, were integrated to simulate high precision hourly PM2.5 concentrations and measure PM2.5 pollution exposure intensity in urban neighborhoods by combining the dynamic population distribution characterized by hourly precision LBS data. On this basis, the spatial pattern of PM2.5 pollution exposure risk is further explored in conjunc-tion with the built environment walkability, revealing the actual PM2.5 pollution exposure risk in the built environment and enriching the understanding of the built environment air pollution exposure risk pattern based on the dynamic distribution of population and PM2.5. This study provides a feasible method and scientific basis for studying the differ-ences in air pollution exposure among different groups and the derived environmental justice issues, and for accurately identifying the high-risk areas and populations of air pollution exposure in urban areas.”(Page 19, Line 652)

Point 2: Some gaps still exist in the present version, specifically in Section 3 and 4, and further improvement and polishment are required for the revised manuscript. Currently, the method section is very vague, please provide some details and remove the useless information.

Response2:Thank you for your suggestion. I have carefully revised the methods section, adding a lot of specific information and removing useless information. You can see from page 6, line 240 to page 9, line 360.

Point 3: In the manuscript, there are many word spelling errors and grammar errors. In addition, many sentences are very wordy and do not provide useful information. Overall, the english language is painful to read. Pls consider polish your language through your coauthors and native english speakers. 

Response3:Thank you for your suggestion. I submitted the article to a professional language polishing agency for a week of polishing and revision.

Point 4: Currently, the quantification of these indicators is very rough. Overall, further details of methods are needed, and an uncertainty analysis of methodological part needs to be conducted in the discussion section. It is not common to use project symbol to summarize results. You used many similar undefined terms, and they are not common. Please check all your manuscript.

Response4:Thank you for your suggestion. I have carefully revised the methods section, adding a lot of specific information and removing useless information. You can see from page 6, line 240 to page 9, line 360. In the discussion section, I discuss the existing deficiencies of the article, which you can see on page 20, line 665.

Reviewer 2 Report

The article is a good work that takes into account the real dynamics of the environment during the daily activity-mobility of an individual by using high-precision air pollution data and LBS-based population spatio-temporal dynamic distribution data and applying geographically weighted regression models, bivariate LISA analysis, and Gini coefficients. However, the overall Merit will be "high" if the following remarks have been done.

1. Using abbreviations in the Abstract is not recommended. It should be done in the main text of the manuscript.

2. Line 137 (OD) should be defined. 

3. Line 142 abbreviation of (MPL) means Mobile-Phone Location not Cell Phone Location.

4. Lines 323:325. Hankou area within the First Ring, the WISCO plant area in the Qingshan District, and the Optics Valley area. These areas are unknown to all readers of this article.

5. Figure 3 6, 8, and 11: The image resolution is low and it reduces the clarity of the image especially the map keys.

6. Line 329 of the manuscript, the text implies that the pollutant exposure intensity of each urban block in Wuhan was determined by the spatial and temporal activities of residents. On the other hand, only spatial activities are depicted on the presented map in figure 4.

7. Figure 7 requires axis titles to be completed as well as the units.

8. Figure, and 10: The title should have its orientation switched so that it reads clockwise for the y axis.

9. The Moran index for children isn’t 0.179 as stated in line 481. Please check it according to Figure 10.

10. In the references section, following references should be corrected, please (4,9,11,18,22,23,27, and 28).

Reviewer 3 Report

1.It is required to give an explanation of the abbreviation before the text.

2.Publications included in the reference must be issued according to the requirements of the journal.

3. Corrections need to be made to the design of the drawings

- legends are not visible in figures 3,6,8,11 (it is better to give a general legend for each figure, and not for its parts)

- in Figure 2 - indicate the 1st, 2nd, 3rd rings and other objects (rivers, industrial facilities, etc.) that you mention in the text (otherwise the description of the result is not clear to readers)

- Fig. 7 - it is necessary to give designations on the abscissa and ordinate axes

- Fig. 11 - it is not indicated which part of the picture refers to children, and which to the group 65+

 5. It is necessary to check the links in the "introduction" section (on lines 76-77 there are only [1], [1])

6. Edit the text (line 480) - the Moran's index in children is incorrectly indicated (in Fig. 10, MI = -0.023, and in the text = 0.179)

7.Clarifications should be made in the "methods" section:

- explain why you chose this particular period for study?

- how many PM2.5 automatic control points are located on the territory of Wuhan and how diversely they cover its area

- for a better understanding of pollution levels and the possibility of comparison with data from other authors, indicate the median and quartiles of PM2.5 concentrations within the 1st, 2nd, 3rd rings.
